# Dynamic Evolutionary Analysis of Land Use/Cover and Ecosystem Service Values on Hainan Island

**DOI:** 10.3390/ijerph20010776

**Published:** 2022-12-31

**Authors:** Zihan Jin, Changsheng Xiong, Qiaolin Luan, Fang Wang

**Affiliations:** 1Department of Land Resources Management, School of Public Administration, Hainan University, No. 58, Renmin Avenue, Haikou 570228, China; 2Department of Economic Management of Agriculture and Forestry, School of Management, Hainan University, No. 58, Renmin Avenue, Haikou 570228, China

**Keywords:** land use, ecosystem service values, intensity analysis, landscape pattern, geodetector

## Abstract

This study of Hainan Island, based on three periods of land use/cover data from 2008, 2013, and 2017, uses the intensity analysis model and landscape pattern index to portray the dynamic changes of land use on the island and a quantitative analysis of the spatial and temporal evolutionary characteristics of ecosystem service values (ESV) based on the equivalent factor method. At the same time, the response of ESV to landscape pattern changes is explored. The results indicate: (1) From 2008 to 2017, the cultivated land in the coastal areas around Hainan Island continued to expand, which squeezed out forest land and reduced its area. The growth of built-up areas in Haikou City and Sanya City was more dramatic. (2) A weakening trend in the intensity of land use on Hainan Island during the study period. There were significant changes in cultivated land, grassland, and bare land, with forest land, grassland, and water bodies transformed into cultivated land. Built-up areas increased mainly through the occupation of cultivated land, grassland, and water bodies. (3) The fragmentation of landscape patches and the diversity of landscapes on Hainan Island increased, with the distribution of landscape types tending to be balanced. (4) From 2008 to 2017, the overall ESV of the island showed an initial decrease before increasing; the main spatial distribution characteristic of the ESV was “high in the central and low in the surroundings”. (5) The mean patch area, the Shannon diversity index, and the largest patch index showed clear positive correlations to ESV.

## 1. Introduction

Since the reform and opening up of China, there has been a period of rapid development, of industrialization, and urbanization, with great achievements in all aspects of the social economy. However, at the same time, these unsystematic patterns of development have led to drastic changes in land use, and the ecological environment on which human beings depend for their survival has been damaged. Against this background, urgent issues for consideration are how to protect and restore the ecosystem, improve the supply of ecosystem services, and ensure the sustainable development of society.

Ecosystem services are the goods and services that humans obtain directly or indirectly from ecosystems, including provisioning, supporting, regulating, and cultural services [1], all of which are highly relevant to human well-being. The estimation of ecosystem service values (ESV) aims to quantify the strength of ecosystem services and is important in helping people understand the status of natural capital, rationalizing future land use, promoting relevant policies, and so on. Therefore, it has received great attention in recent years and has become a research hotspot in ecology, geography, and other disciplines. Among the estimation methods, the equivalence factor has been widely used by scholars at home and abroad as it requires less data, is straightforward to operate and implement, and uses more comprehensive and comparable assessment objects [2,3,4]. In 1997, Costanza et al. [5] established the ESV assessment model in *Nature*, and Xie et al. [6] modified the coefficients for the specific situation in China to produce the “Table of Equivalent Factors of Terrestrial Ecosystem Service Value in China”. Since then, a large number of domestic scholars have made reference to the equivalent factor table measured by Xie Gaodi, revised it according to the actual local conditions, and achieved rich research results in the evaluation of ecosystem service values in many areas in China, such as ecologically fragile areas in the west [7,8], important river basins [9,10], urban agglomerations [11,12,13] and key ecological function areas [14,15,16].

Land cover change intensity analysis is an analytical method proposed by Professor Pontius’ research group at Clark University, USA. Based on the land cover transformation matrix, it can comprehensively and systematically reveal the internal linkage of land cover change processes and patterns from the two perspectives of transfer-in and transfer-out [17,18,19]. Compared to other common land use and land cover change (LUCC) analysis models, it is a quantitative, mathematical, and theoretical framework that can systematically quantify the internal transformation status in the dynamic land use change process and examine the intensity changes at time intervals and by land use types and transfer levels. Huang et al. [20] applied the method for the first time to the Jiulong River Basin in southeastern China to explore the region’s land use change process and patterns. Yang et al. [21] used a cross-linked table of transformation patterns in an analysis of land use intensity in Wuhan City to demonstrate the stability and systematic characteristics of the transformation of each land use type. Deng et al. [22] used this method to analyze the evolution characteristics of production-living-ecological spaces in Hengyang City.

Landscape pattern can be understood as the characteristics of land use/cover in spatial distribution, which can reflect the behavior of potential human activities. It has gradually become one of the main analytical tools for land use/cover change research. At present, the studies of landscape pattern evolution and the response of ESV under land use change have been relatively rich and deep, but the traditional LUCC analysis models are still widely used [23,24]. In most LUCC analyses using the intensity analysis model, very few studies also consider the interaction between landscape pattern and regional ESV [10]. Therefore, it is important to make use of the advantages of intensity analysis models to compensate for the shortcomings of other LUCC analysis models before conducting an in-depth exploration of the response relationship with landscape pattern and ESV.

In 2008, the government of Hainan Province officially launched the International Tourism Island Construction Plan, and under its influence, land use/cover and ESV in the region changed significantly. In order to rule out any possible impact of the 2018 Hainan Free Trade Port Strategy, this study focuses on the changes of land use/cover and ESV on Hainan Island during the international tourism island construction period between 2008 and 2017. Based on the land use/cover data of the three phases of 2008, 2013, and 2017, the intensity analysis model and the landscape pattern indices were used to jointly depict the dynamic changes of land use on the island. Subsequently, based on the equivalence factor method, the temporal and spatial evolution characteristics of the ESV were analyzed, and the relationship between the landscape pattern and the ESV was explored. The purpose of this article is to fill the research gap in the overall land use change and ecosystem service values assessment of Hainan Island under the background of a specific development strategy, help people better recognize and understand the role of local landscape pattern on ecosystem service functions, and provide reference for the decision-making of ecological management and optimization.

## 2. Study Area and Data Sources

### 2.1. Overview of the Study Area

Hainan Province, consisting of the Hainan, Nansha, Zhongsha, and Xisha islands, is located at the southernmost tip of China and is separated from Guangdong Province by the Qiongzhou Strait in the north. The total land area of the province is 35,400 square kilometers, of which Hainan Island is about 33,900 square kilometers. It is known as the second largest island in China after Taiwan. The terrain of Hainan Island is low and flat around, and high in the middle. It has a tropical monsoon climate with abundant precipitation, a year-round stable forest cover of over 60%, a wide variety of plants, and one of the highest quality ecological environments in China. Due to the combination of natural and transportation conditions and historical development, its main population and socio-economic activities are mostly distributed in the eastern coastal areas, Haikou City in the north, and Sanya City in the south. As Sansha City functions as the center of national defense, this article delineates the study area to be 18 cities and counties in Hainan Island (Figure 1).

### 2.2. Data Sources

The land use/cover data of Hainan Island for 2008, 2013, and 2017 and the national county-level administrative boundary vector data were obtained from the *Resource and Environment Science Data Center of the Chinese Academy of Sciences* (http://www.resdc.cn/, accessed on 1 November 2021. The former had a spatial resolution of 30 m. With reference to *GB/T 2010–2017 “Status of Land Use Classification,”* based on ArcGIS 10.8 software, the land use/cover was reclassified into six categories: cultivated land, forest, grassland, water bodies, bare land, and built-up area, and the county-level boundary vector data were uniformly projected to WGS_1984_UTM_Zone_49N. In addition, the statistical data of grain output value, grain price, Engel’s coefficient, and the urban-rural population ratio used in the calculation of ESV for each year were obtained from the China Statistical Yearbook, the Statistical Yearbook of Hainan Province, the Statistical Bulletin of National Economic and Social Development of Hainan Province, and relevant government websites.

## 3. Research Methods

### 3.1. Intensity Analysis

The intensity analysis model expresses the intensity of change of each land use type in different time periods in a bottom-up, cascading, quantitative manner using three levels: the time interval level, the feature type level, and the transfer level [25].

#### 3.1.1. Time Interval Level

Interval-level intensity analysis can reflect the overall land use change within each time interval by calculating the annual average land use change intensity *S_t_* for each time interval and comparing it with the total change intensity *U* for the whole study period to derive whether the rate of change is fast or not. If the land use change within the time interval is rapid, then *S_t_* > *U;* if the land use change within the time interval is slow, then *S_t_* < *U*; if the land use change shows absolute stability within the study period, then *S_t_* = *U*. The formula for the time interval level is:(1)$$S_{t}=\frac{\sum_{j=1}^{J}\left[\left(\sum_{i=1}^{J}C_{tij}\right)-C_{tij}\right]}{\left(Y_{t+1}-Y_{t}\right)\left(\sum_{j=1}^{J}\sum_{i=1}^{J}C_{tij}\right)}\;\times 100\%$$
(2)$$U=\frac{\sum_{t=1}^{T-1}\left\{\left(Y_{t+1}-Y_{t}\right)\sum_{j=1}^{J}\left[\left(\sum_{i=1}^{J}C_{tij}\right)-C_{tij}\right]\right\}}{\left(Y_{T}-Y_{1}\right)\sum_{t=1}^{T-1}\left[\left(Y_{t+1}-Y_{t}\right)\left(\sum_{j=1}^{J}\sum_{i=1}^{J}C_{tij}\right)\right]}\;\times 100\%$$
where *i* is the land use type at the initial time node of the time interval; *j* is the land use type at the final time node of the time interval; *J* is the number of land use types; *t* is the time nodes on the time interval $\left[Y_{t},Y_{t+1}\right]$; *T* is the total number of time nodes;$\;Y_{t}$ is the year corresponding to time node t; $C_{tij}$ is the number of elements transferred from land type *i* to land type *j* in the time interval $\left[Y_{t},Y_{t+1}\right]$, i.e., the increase in land type *j*; $S_{t}$ is the average annual intensity of change in the time interval $\left[Y_{t},Y_{t+1}\right]$; and *U* is the total intensity of land use change in the whole study period.

#### 3.1.2. Feature Type Level

Feature-type-level intensity analysis is based on the change area of each land use type, calculating and analyzing the increase or decrease in area of each land use type and the corresponding intensity change magnitude, and comparing this with the annual average change intensity to derive whether the increase or decrease in area change of each land use type is active or not. If the intensity change in the increase or decrease of land use types in a certain time period is active, then *L_ti_* > *S_t_* or *G_tj_* > *S_t_*; if it is relatively stable, then *L_ti_* < *S_t_* or *G_tj_* < *S_t_*; if it is absolutely stable, then *L_ti_ =S_t_* or *G_tj_ S_t_*. The formula for feature type level is:(3)$$L_{ti}=\frac{\left[\left(\sum_{j=1}^{J}C_{tij}\right)-C_{tii}\right]/\left(Y_{t+1}-Y_{t}\right)}{\sum_{j=1}^{J}C_{tij}}\;\times 100\%$$
(4)$$G_{tj}=\frac{\left[\left(\sum_{i=1}^{J}C_{tij}\right)-C_{tjj}\right]/\left(Y_{t+1}-Y_{t}\right)}{\sum_{i=1}^{J}C_{tij}}\;\times 100\%$$
where $C_{tii}$ is the area that shifted from land use type *i* at the beginning of the period to land use type *i* at the end of the period in the time interval $\left[Y_{t},Y_{t+1}\right]$, i.e., the area that has not changed; $C_{tjj}$ is the area transferred from land use type *j* at the beginning of the period to land use type *j* at the end in the time interval $\left[Y_{t},Y_{t+1}\right]$; $L_{ti}$ is the average annual loss intensity of land use type *i* in the time interval $\left[Y_{t},Y_{t+1}\right]$ relative to time *t;*
$G_{tj}$ is the average annual increase intensity of land use type *j* in the time interval $\left[Y_{t},Y_{t+1}\right]$ with respect to time *t* + 1.

#### 3.1.3. Transfer Level

Transfer-level intensity analysis can reflect the direction and degree of the conversion between different land use types in a given time interval. When the transfer intensity *R_tin_* > average increase intensity *W_tn_,* this indicates that the increase of land use type *n* may originate from the occupation of land use type *i.* When *R_tin_* < *W_tn_*, it indicates that the increase of land use type *n* avoids the occupation of land use type *i.* When the transfer intensity *Q_tmj_* > average increase intensity *V_tm_,* it indicates that the decrease of land use type *m* may transform into land use type *j*. When *Q_tmj_* < *V_tm_*, it indicates that the reduction of land use type *m* is not transformed into land use type *j*. The formula for the transfer level is:(5)$$R_{tin}=\frac{C_{tin}/\left(Y_{t+1}-Y_{t}\right)}{\sum_{j=1}^{J}C_{tij}}\;\times 100\%$$
(6)$$W_{tn}=\frac{\left[\left(\sum_{i=1}^{J}C_{tij}\right)-C_{tnn}\right]/\left(Y_{t+1}-Y_{t}\right)}{\sum_{j=1}^{J}\left[\left(\sum_{i=1}^{J}C_{tij}\right)-C_{tnj}\right]}\;\times 100\%$$
(7)$$Q_{tmj}=\frac{C_{tmj}/\left(Y_{t+1}-Y_{t}\right)}{\sum_{i=1}^{J}C_{tij}}\;\times 100\%$$
(8)$$V_{tm}=\frac{\left[\left(\sum_{j=1}^{J}C_{tmj}\right)-C_{tmm}\right]/\left(Y_{t+1}-Y_{t}\right)}{\sum_{i=1}^{J}\left[\left(\sum_{j=1}^{J}C_{tij}\right)-C_{tim}\right]}\;\times 100\%$$
where $C_{tin}$ is the area of land use type *n* that shifted from land use type *i* at the beginning of the period to land use type *n* at the end in the time interval $\left[Y_{t},Y_{t+1}\right]$; $C_{tnn}$ is the area of land use type *n* that does not change in land use type during the study time interval; $R_{tin}$ is the average annual transfer-in intensity of land use type *i* transferring to land use type *n* during the time interval $\left[Y_{t},Y_{t+1}\right]$ relative to time *t* (*i ≠ n*); $W_{tn}$ is the average annual equilibrium transfer intensity of all *non-n* land use types transferring to land use type *n* during the time interval $\left[Y_{t},Y_{t+1}\right]$ relative to time *t*; $C_{tmj}$ is the area transferred from land use type *m* at the beginning of the period to land use type *j* at the end in the time interval $\left[Y_{t},Y_{t+1}\right]$; $C_{tmm}$ is the area of land use type *m* that does not change in land use type during the study time interval; $Q_{tmj}$ is the average annual transfer-out intensity of land use type *m* transferring to land use type *j* in the time interval $\left[Y_{t},Y_{t+1}\right]$ relative to land use type *j* at time *t* + 1 (*j* ≠ *m*); $V_{tm}$ is the average annual equilibrium transfer intensity of land use type *m* transferring to *non-m* land use types in $\left[Y_{t},Y_{t+1}\right]$ relative to all *non-m* land use types at time *t* + 1.

This study borrowed ideas from the cross-linked table of two land conversion patterns of area increase and decrease pioneered by Wang et al. [10]. As shown in Figure 2, each horizontal row of ① and ②, respectively, represents the two time intervals for 2008–2013 and 2013–2017, and the color represents whether the land use type *j* tends to or avoids conversion to land use type *i*. The light green, indicating that the conversion of land use type *j* to land use type *i* shows low avoidance in the corresponding time interval, while dark green indicates high avoidance. Similarly, yellow and red, respectively, represent a low or high tendency of land use type *j* to be transformed into land use type *i*. The colorless part has two meanings: one is that its corresponding land use types *i* and *j* are the same, that is, this part of the land has not undergone land use type transformation. The other is that its corresponding land use types *i* and *j* are not the same, but the area converted from land use types *j* to *i* is 0 in the corresponding time interval.

### 3.2. Landscape Pattern Analysis

The landscape index is a quantitative indicator of the characteristics of the structural composition and spatial configuration of the landscape pattern. It establishes the connection between landscape structure and the process or phenomenon and can better explain landscape function [26]. In order to comprehensively reflect the landscape pattern characteristics of Hainan Island, the number of patches (NP), mean patch area (AREA_MN), Shannon diversity index (SHDI), Shannon evenness index (SHEI), aggregation index (AI), and spread degree (CONTAG) were selected at the landscape level. The patch density (PD), mean patch area (AREA_MN), landscape shape index (LSI), the largest patch index (LPI), and the interspersion and juxtaposition index (IJI) were selected at the level of landscape type with reference to the integration of the indicators selected in previous studies. The above indexes can reflect the shape, area, fragmentation, diversity, aggregation, and dispersion characteristics of the landscape of the study area in a comprehensive manner.

### 3.3. Ecosystem Service Values Assessment

Based on the standard equivalent factor table proposed by Xie [6], the assessment model is as follows:(9)$$ESV=\sum_{i=1}^{i}\sum_{j=1}^{j}A_{i}VC_{ij}S_{t}$$
where: *ESV* is the total ecosystem service values; i is the land use type; j is the function of the ecosystem service; *A_i_* is the area of the land use type *i;* $VC_{ij}$ is the unit area value equivalent factor (yuan/hm^2^) of the ecological service function *j* provided by land use type *I* (i.e., the product of the single service value factor and a standard equivalent factor); and *S_t_* is the social development correction coefficient in year *t.*

A standard ecological service value equivalent factor (referred to as the standard equivalent factor) is determined according to 1/7 of the economic value of food production services provided by unit area farmland [27]. On this basis, in order to exclude the effects through inflation or contraction and make the economic value of food comparable between years, this study uses the year-by-year consumer price index of Hainan Province to correct this, and the formula obtained is as follows:(10)$$E_{t}=\frac{1}{7}\frac{U_{t}}{M_{t}}$$
(11)$$U_{t}=U_{t}^{\prime }\sum_{t+1}^{t=2017}a_{t}$$
(12)$$a_{t}=\frac{CPI}{100}$$
where $E_{t}$ is the economic value of one standard equivalent factor on Hainan Island in year t. $U_{t}$ is the constant value obtained by converting the total food production value of Hainan Island in year *t* to the base year of 2017. $M_{t}$ is the area under grain cultivation in Hainan Island in year *t.*
$U_{t}^{\prime }$ is the actual total food production value for Hainan Island in year *t*; and *CPI* is the consumer price index. The economic value of one standard equivalent factor of Hainan Island at the three research time points is calculated as 1924.31 yuan/hm^2^, 1807.26 yuan/hm^2^ and 1974.21 yuan/hm^2^.

It should be noted that the magnitude of the ESV is closely related to the stage of social development, which also influences people’s willingness to pay for ecosystem functions and services [28,29]. In this study, the correction coefficient of social development is determined according to the Peel growth curve model, and the specific formulae are as follows:(13)$$Q=\frac{1}{1+ae^{-b(\frac{1}{E_{n}}-3)}}$$
(14)$$Q^{\prime }=Q_{1}\times P_{1}+Q_{2}\times P_{2}\;$$
(15)$$S_{t}=\frac{{Q^{\prime }}_{study\;area}}{{Q^{\prime }}_{country}}\;$$
where: *Q is* the coefficient of social development related to willingness to pay; generally constants *a* and *b* are 1; *E_n_* is the Engel coefficient; and *e* is the natural constant; $Q_{1}$ and $Q_{2}$ respectively denote the urban and rural social development coefficient; $P_{1}$ and $P_{2}$ respectively denote the urban and rural population in the total population; ${Q^{\prime }}_{study\;area}$ and ${Q^{\prime }}_{country}$ are the comprehensive social development coefficients of Hainan Island and the whole nation; *S_t_* is the social development correction coefficient in year *t.*

Based on the actual land use types on Hainan Island, the coefficients of paddy fields and broad-leaved forests in the equivalent table are taken for cultivated land and forest in the calculation process. Grassland takes shrub’s coefficient, and the coefficients of water systems and wasteland are taken for water bodies and bare land.

### 3.4. Geodetector

Geodetector is a set of statistical methods for detecting spatial heterogeneity and the driving forces behind it. The magnitude of the explanatory power (i.e., q-value) of each driver on the distribution of the dependent variable can be detected quantitatively through spatial heterogeneity on the basis of almost no assumptions [30]. In this study, the response of city- and county-scale ESV to changes in landscape patterns was analyzed using the geodetector model with the following equation:(16)$$q=1-\frac{\sum_{h=1}^{L}N_{h}\sigma_{h}^{2}}{N\sigma^{2}}$$
where: *q* is the magnitude of the explanatory power of the potential drivers (i.e., the 10 landscape pattern indices selected in this study) on the dependent variable (i.e., the value of ecosystem services in each city and county), and the larger the *q* value, the greater the influence of the drivers on the dependent variable and vice versa; *L* denotes the stratification of the drivers and the dependent variable, which in this study specifically refers to each city and county on Hainan Island; *N* and $N_{h}$ are respectively the number of samples in the whole area and sub-region of the study area; $\sigma^{2}$ and $\sigma_{h}^{2}$ denote the variance of the whole region and sub-region.

## 4. Results and Analysis

### 4.1. Analysis of Land Use Change

#### 4.1.1. Spatial Change in Land Use

From 2008 to 2017, the overall land use change on Hainan Island showed a trend of cultivated land and built-up area increasing and forest land decreasing. As can be seen from Figure 3, the expansion of cultivated land in the surrounding coastal areas crowded out forest land and reduced its area, especially in the north. The built-up area showed a slight increase overall, but the growth was relatively dramatic in Haikou City in the north and Sanya City in the south. The area of the remaining three types did not change significantly during the study period.

#### 4.1.2. Analysis of the Intensity of Land Use Change

##### Intensity Analysis at the Time Interval Level

The change area and intensity of land use at the time interval level on Hainan Island during the study period both showed a decrease. The change area at the interval from 2008–2013 was significantly higher than that from 2013–2017, at 8.99% and 6.93%, respectively (left half of Figure 4). The average change intensity of land use in the two-time intervals showed a slight decreasing trend (right half of Figure 4), where the intensity of land use change in 2008–2013 was higher than the average intensity of the whole study period, indicating that the change in land use intensity during this time interval was rapid. The intensity of land use change in 2013–2017 was lower than the average intensity of the whole study period, thus showing a slow change.

##### Intensity Analysis at the Feature Type Level

In this study, the increase in cultivated land and the decrease in forest land were the highest area changes, while the smallest area change was seen in bare land, which was almost negligible (left half in Figure 5 and Figure 6). The difference between the transfer in and out of cultivated land and forest from 2008–2013 was not significant; however, the decrease in cultivated land and the increase in forest land during 2013–2017 slowed rapidly, resulting in a significant increase in the area of cultivated land and a decrease in the area of forest. The change intensity in both the increase and decrease of cultivated land, grassland, and bare land during the study period was consistently greater than the average change intensity, indicating that they were all more active. In contrast, the increase and decrease in the change intensity of forest land during the same period were lower than the average, with the amount of forest remaining relatively stable. Although the average annual increase and decrease in area of forest land were the highest, the intensity of change was not significant due to its large base figure. In addition, the increase in intensity of the built-up area was higher than the average intensity, however, the decrease in intensity was much smaller than average, reflecting its relatively active state.

##### Intensity Analysis of the Transfer Level

The transformation of land use types on Hainan Island during the study period was substantial (Figure 7 and Figure 8). From 2008 to 2017, the transformation between different land use types was mainly from cultivated land and water bodies to built-up areas and the mutual transformation between grassland, water bodies, and cultivated land. Most of the increase in cultivated land area was due to the occupation of grassland and water bodies, and the increase in the built-up area was mainly due to the decrease in cultivated land, grassland, water bodies, and bare land. In contrast, the area of forest was more stable, with little change in increase and decrease as a result of its large area base.

### 4.2. Analysis of the Evolution of Landscape Pattern

#### 4.2.1. Landscape Level Analysis

As can be seen from Figure 9, trends in the number of patches and the mean patch area of Hainan Island reversed during the study period, with the former showing a decline followed by a rise and the latter a rise followed by a decline. Both the spread degree and aggregation index characterize the aggregation status of different landscape types. These showed a decreasing trend overall, indicating that the aggregation of landscape patches in Hainan Island continued to weaken and the fragmentation degree increased. The trends of the Shannon diversity index and Shannon evenness index were more or less the same, both showing a continuous increase, indicating that the landscape diversity in the study area increased, and that the distribution of each landscape type tended to be balanced.

#### 4.2.2. Landscape Type Level Analysis

By analyzing the landscape indices of different land use patch types (Figure 10), it was seen that the patch density of forest was gradually increasing, while its mean patch area was gradually decreasing; this showed that the patch fragmentation degree of the forest landscape was continuously aggravating. Meanwhile, the largest patch index showed that the forest landscape was dominant in the study area, although its position was gradually declining. This might be due to the fact that, since the construction of the international tourism island, major projects have been launched, resulting in the dramatic increase of demand for land. Thus, the intensity of human exploitation activities on forest land saw a notable increase, with many large patches divided into smaller ones, leading to an increase in the fragmentation degree of forest patches. The landscape shape index of cultivated land was significantly higher than that of other land use types and showed an increasing trend overall, which indicated that the patch shape of cultivated land was becoming more complex. From 2008 to 2017, the interspersion and juxtaposition indices of each landscape type showed an increasing trend, indicating that the richness of each neighboring landscape type increased.

### 4.3. Analysis of Dynamic Changes in Ecosystem Service Values

#### 4.3.1. Spatial Variation

In this paper, a 30 m × 30 m raster was used as the basic research unit to explore the spatial variation pattern of ESV on Hainan Island, and we used the Natural Jecks method to classify the ESV into 5 grades: low, relatively low, medium, relatively high, and high in ArcGIS 10.8 software, according to the three periods of ESV on Hainan Island. Finally, we got the spatial distribution of ESV grades on Hainan Island for 2008, 2013, and 2017 (Figure 11).

From Figure 11, the basic spatial distribution pattern of ESV grades on Hainan Island was relatively stable, showing the overall spatial distribution characteristic of “high in the middle and low in the surroundings”. Medium and relatively high ESV-grade areas accounted for the largest proportion. The relatively high-grade areas were mainly located in the mountainous areas with extensive forests in the central part, while the medium grade areas were concentrated in the coastal areas around Hainan Island. The low- and high-grade areas were scattered, with the former being most concentrated in Haikou City in the north and Sanya City in the south, while the latter was mainly distributed in the south of Danzhou City, the east of Dongfang City, and the west of Ledong City. The spatial variation of ESV on Hainan Island during the study period corresponded to the spatial variation of local land use. The corresponding ESV coefficients of cultivated land are lower than those of forest land, so the encroachment of cultivated land on forest land within 2008–2017 made the scope of relatively high-grade areas gradually shrink inward. Meanwhile, the range of medium-grade areas continuously expanded, which was most evident in Chengmai City, Haikou City, and Wenchang City in the north.

#### 4.3.2. Time Variation

Analysis of the time scale showed that the overall ESV of Hainan Island initially decreased and then increased in general, with a lower ESV at the end of the study period than at the beginning. Regulation and support services have always been the core functions of ecosystem services on Hainan Island (Table 1).

Specifically, among the primary services, the only one showing negative values was the water supply service. The reason for this was that the coefficient of water supply service for cultivated land was 2.63, gradually increasing in quantity, especially in the period from 2013 to 2017. The proportion of water bodies that had the highest coefficient of water supply service was much lower than that of cultivated land and fluctuated during the study period. As a result, the values of the service decreased much more than they increased, always presenting negative values and showing a high negative growth rate between 2013 and 2017. The proportion of values generated by all primary types of services changed slightly, indicating that various ecosystem services on Hainan Island were relatively stable during this period. The proportion of regulation services always maintained a level above 70%, followed by support services, while cultural and supply services had smaller proportions. From the analysis of numerical characteristics, the change trend of overall values of the four service types was consistent, all initially showing a decline before increasing. The values at the end of the period were lower than at the beginning, with the highest value of all types of services recorded in 2008 and the lowest value in 2013.

Forest land contributed more ESV than other land use types, followed by water bodies (Table 2). The contribution of grassland and bare land was almost negligible. From 2008 to 2013, changes in forest land led to the largest decrease in ESV (−25.350 billion yuan), followed by water bodies (−3.187 billion yuan) and cultivated land (−1.508 billion yuan); changes in other land use types led to a weak decreasing trend in ESV. From 2013 to 2017, the ESV contributed by most land use types showed a rebounding trend, with the most significant growth trend of ESV for forest land (13.314 billion yuan), and only the ESV contributed by grassland still presenting negative growth during this period. Between 2008 and 2017, except for cultivated land, the ESV of all other land use types presented negative growth, which was related to the steady increase in the area of cultivated land.

### 4.4. Relationship between Landscape Pattern and Ecosystem Service Values

Results from the geodetector showed a variability in the degree of influence of different landscape pattern indices on the changes in the ESV of Hainan Island (Table 3). It was seen that the q-values of four factors, NP (number of patches), PD (patch density), CONTAG (spread degree), and AI (aggregation degree), were all less than 0.1, indicating their weak explanatory effects on the dependent variable. Among the remaining six factors, the ranking of the influence degree of each factor on ESV, in descending order, was AREA_MN (mean patch area) > SHDI (Shannon diversity index) > LPI (largest patch index) > IJI (interspersion and juxtaposition index) > SHEI (Shannon evenness index) > LSI (landscape shape index). The q-values of these driving factors were all greater than 0.1, having a certain explanatory effect, with the q-values of AREA_MN, SHDI, and LPI being significantly higher than the others, indicating that these had stronger influences on the regional ESV. Consequently, the increase in area of each patch, the tendency of the landscape to diversity, and the increase in the area of dominant landscape patches in the study area would contribute to an increase in the ESV of the island. From the significance results, the p-values of all three factors passed the significance test of 0.05, indicating that the detection result was relatively reliable.

## 5. Discussion

### 5.1. Analysis of the Causes of Land Use Intensity Change

On the time interval level, the overall intensity of land use change in Hainan Island from 2013–2017 was slightly lower than that of 2008–2013, when the intensity of increase in built-up areas showed the same downward trend as the overall intensity. The reason was that the study period coincided with the height of the international tourism island construction, when Hainan was making great efforts to improve the quality of its tourism services, to create a tourism industry system with regional characteristics, and to reach an internationally advanced level. Within this development context, in order to improve tourism-related facilities to meet the high-quality needs of future domestic and foreign tourists, it was inevitable that some ecological land would be part of this development, resulting in significant changes in the nature of the surface features. The first half of the study period, 2008–2013 was also the early stage of construction of the international tourism island. At this time, multiple favorable policies were implemented as the development of tourism accelerated. The need for coastal resorts, rural tourism, characteristic towns, and related cultural and creative industries, meant various large-scale facilities supporting tourism were built on the island, leading to an initial rapid change in land use. In later stages of development, the average annual growth rate of tourist traffic lagged behind that of the tourism industry, and the number of these facilities proved to be excessive. As a result, the pace of construction of the international tourism island subsided, the intensity of development activities weakened, and the rate of land use change showed a corresponding decline.

The results at the feature type level suggest that the significant increase in cultivated land and the decrease in forest land between 2013 and 2017 were caused by the growing national emphasis on food security that occurred after the 18th National Congress. With greater protection of cultivated land, local governments across the country issued a series of policy documents to implement the strictest cultivated land protection system, and Hainan Province was no exception. As the responsibility of local governments for cultivated land protection grew, in order to complete the appraisal tasks of obligatory targets, they inevitably ignored the protection of other ecological land. Hainan was an ecological province with rich forest resources, but due to insufficient attention paid to them, deforestation for cultivated land reclamation and encroachment on forests gradually decreased their numbers.

### 5.2. Response of Ecosystem Service Values to Changes in Landscape Pattern

Land use/cover change caused changes in landscape pattern, which in turn affected ecosystem material cycles and energy flows, resulting in changes in ESV in the study area. Therefore, in order to better guide the sustainable development of regional ESV, it is necessary to consider the evolution of landscape patterns in addition to the quantitative changes in regional land use types. Cen Xiaoteng [31] analyzed the relationship between ESV and landscape pattern in the south coast area of Hangzhou Bay and found that the richer the land use, the more fragmented the landscape, and the higher the degree of diversity, which is conducive to improving the overall service values. Zheng et al. [32] took Gannan as an example to study the impact of landscape pattern changes on ESV and concluded that total ESV showed a strong positive correlation with CONTAG and was negatively correlated with SHDI. Zhang et al. [33] analyzed the response of ESV to the evolution of the landscape pattern in Shishou City and found that ESV was significantly positively correlated with AREA_MN, SHDI, and NP. In this paper, it is found that AREA_MN, SHDI, and LPI have a large impact on regional ESV, indicating that high contiguity of patches and diversification of landscape types may have positive effects on ecosystem service values, which is basically consistent with Zhang et al. According to many scholars’ research, different research time points, scale effects, and regional differences may make the impact of landscape pattern indices on ecosystem service values heterogeneous [34,35,36].

### 5.3. Related Policy Recommendations

In order to further improve the supply of ecosystem services on Hainan Island, it is necessary to plan and manage the natural resources and human activities in a scientific and rational way. On the one hand, based on the major influences of both the mean patch area and largest patch index on the ESV, it is recommended to enhance the concentration and contiguity of cultivated land by reclaiming scattered, inefficient construction land between fields and organizing agricultural land. At the same time, it is also necessary to pay attention to the construction of nature reserves so they continue to exert an ecological effect. Hainan is an ecological province with a large number of nature reserves over a vast area, but deforestation for cultivated land reclamation and encroachment on forests and wetlands are still serious issues. We expect that the local government will soon issue provincial guidelines for the management of nature reserves to strictly control land development and utilization activities within the scope of nature reserves, curbing landscape fragmentation and ecological benefit loss. On the other hand, the diversification of landscape types will also have a positive effect on ESV; therefore, governments ought to establish a holistic view of the ecosystem and promote integrated management of “mountain, water, forest, field, lake and grass”. A systematic approach is needed, with coordinated protection of all kinds of ecological land. In this way, an annual increase in ESV is possible by maintaining stability to achieve the future coordinated and sustainable development of the social, economic, and ecological environments on Hainan Island.

## 6. Conclusions

Taking Hainan Island as the research object, based on three periods of land use/cover data from 2008, 2013, and 2017, this study used the intensity analysis model and landscape pattern index to jointly portray the dynamic changes of land use on the island, followed by quantitative analysis of the spatial and temporal evolution characteristics of ESV based on the equivalent factor method. Simultaneously, the response of ESV to landscape pattern changes was explored in order to provide reference for the decision-making of ecological management and optimization. This article offers the following conclusions:(1)From 2008 to 2017, the overall land use variation on Hainan Island showed a trend of cultivated land and built-up area increasing and forest land decreasing. The cultivated land in the surrounding coastal areas continued to expand, crowding out forest land and reducing its area. The growth of built-up areas in Haikou City and Sanya City was more dramatic.(2)The intensity of land use change on Hainan Island during the study period showed a weakening trend. Changes in the area of cultivated land, grassland, and bare land were significant, with the increase in cultivated land mainly caused by the transformation of forest land, grassland, and water bodies. The increase in the built-up area was mainly due to the occupation of cultivated land, grassland, and water bodies.(3)The fragmentation of landscape patches and the diversity of landscapes on Hainan Island increased, with the distribution of landscape types tending to be balanced. From the landscape type level, the fragmentation of forest landscape patches was increasing, and the patch shape of cultivated land was becoming increasingly complex.(4)From 2008 to 2017, the overall ecosystem service values of Hainan Island showed a trend of first decreasing and then increasing, with regulation and support services being the core functions. The main spatial distribution characteristic of ESV on Hainan Island was “high in the central and low in the surroundings”.(5)The mean patch area, Shannon diversity index, and largest patch index had a significant positive relationship with ESV, indicating that the increase in area of each patch, the tendency of the landscape to diversity, and the increase if the area of dominant landscape patches would contribute to the increase of the ESV on Hainan Island.

The following shortcomings of this study should be noted. First, the limited number of time intervals causes almost no color change in the horizontal adjacent cells in the transfer level intensity analysis; therefore, it is impossible to further assess the long term stability of the transition from land use type *i* to land use type *j*. Second, the various service coefficients of the built-up area are classified as zero when accounting for ESV, without considering its potential negative impacts on regional ESV as ecosystem consumers. It is planned to use the indirect market approach to effectively characterize the negative ESV brought by the “three wastes” in further studies. Third, we do not consider the spatial heterogeneity of ESV among “different fields of the same land use type”. It is intended to refine the ESV estimation method by establishing a system of correction factors, including NPP, greenness, humidity, and other indicators, in order to provide a methodological reference for future GEP accounting in the region.

## Figures and Tables

**Figure 1 ijerph-20-00776-f001:**
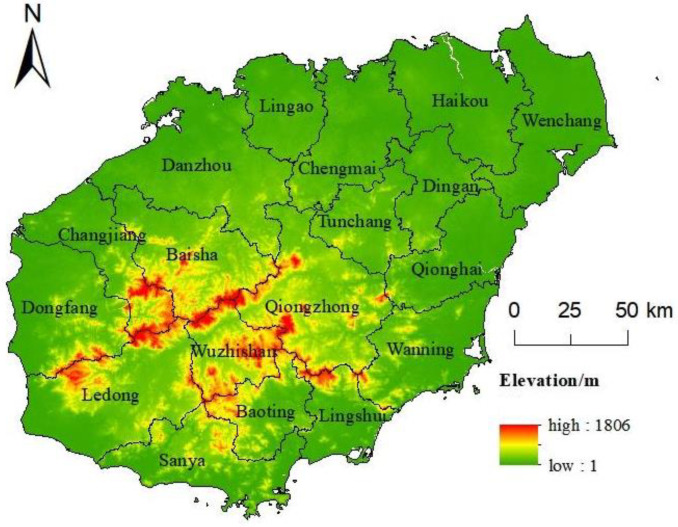
Overview of the study area.

**Figure 2 ijerph-20-00776-f002:**
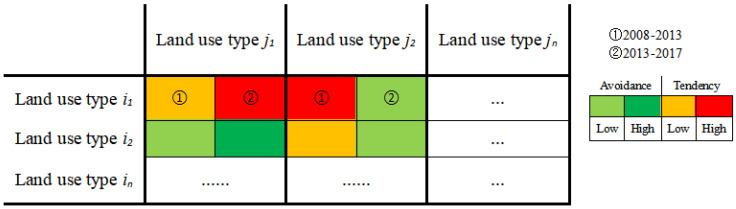
Cross-linked table of land use conversion.

**Figure 3 ijerph-20-00776-f003:**
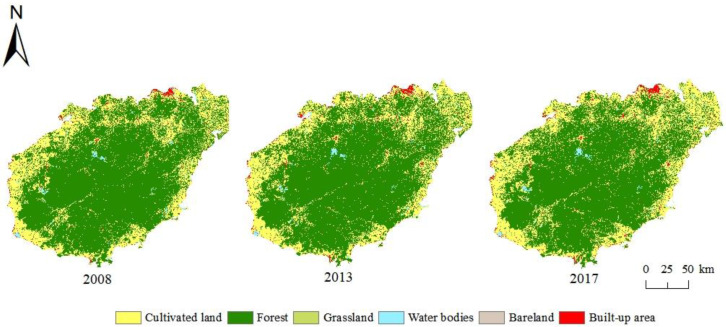
Spatial change of land use on Hainan Island from 2008 to 2017.

**Figure 4 ijerph-20-00776-f004:**
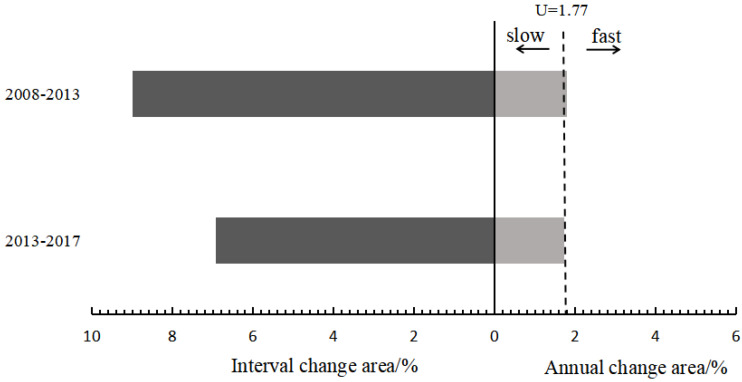
Intensity of land use change at time interval level on Hainan Island, 2008–2013.

**Figure 5 ijerph-20-00776-f005:**
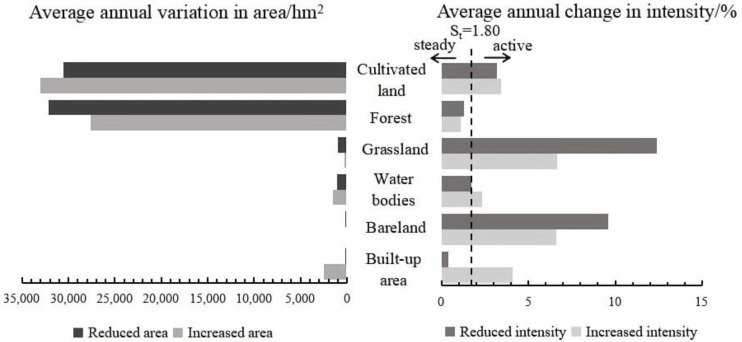
Intensity of land use change at the feature type level on Hainan Island, 2008–2013.

**Figure 6 ijerph-20-00776-f006:**
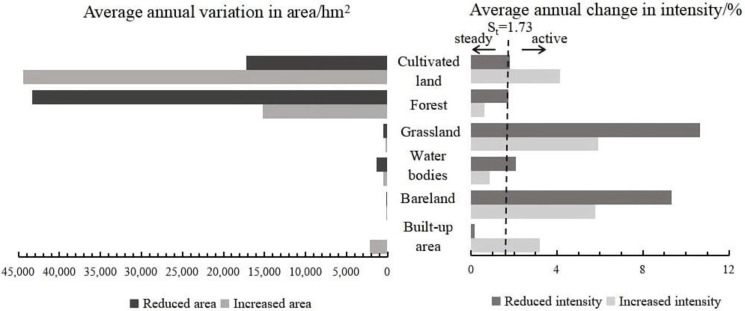
Intensity of land use change at the feature type level on Hainan Island, 2013–2017.

**Figure 7 ijerph-20-00776-f007:**
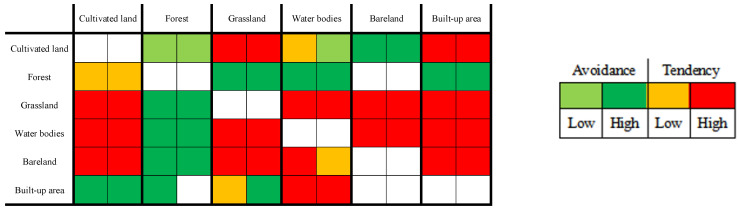
Cross-linked table of land use change patterns at the transfer level on Hainan Island—increase in land area, 2008–2017.

**Figure 8 ijerph-20-00776-f008:**
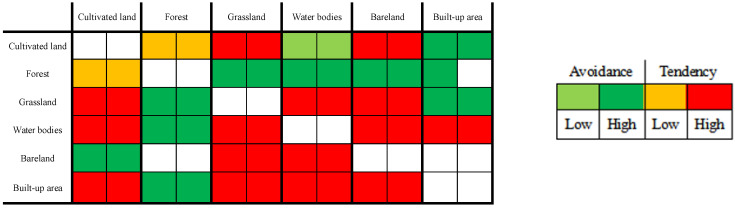
Cross-linked table of land use change patterns at the transfer level on Hainan Island—decrease in land area, 2008–2017.

**Figure 9 ijerph-20-00776-f009:**
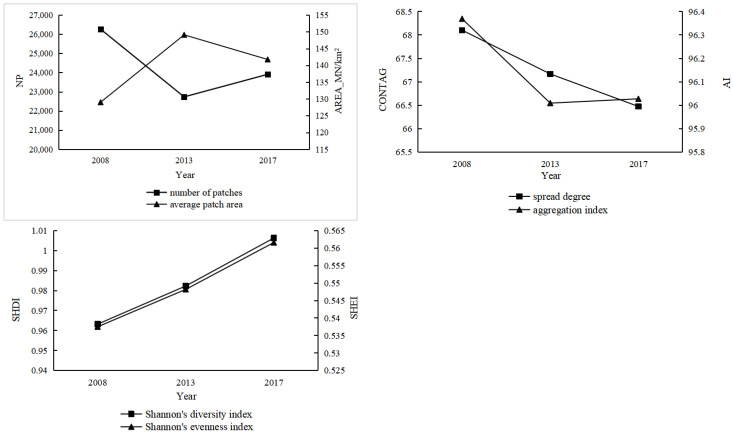
Change in landscape level index of Hainan Island, 2008–2017.

**Figure 10 ijerph-20-00776-f010:**
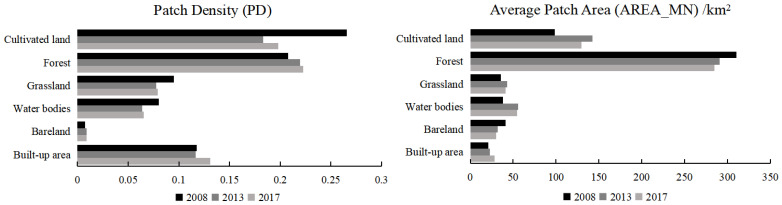
Change in landscape type level index of Hainan Island, 2008–2017.

**Figure 11 ijerph-20-00776-f011:**
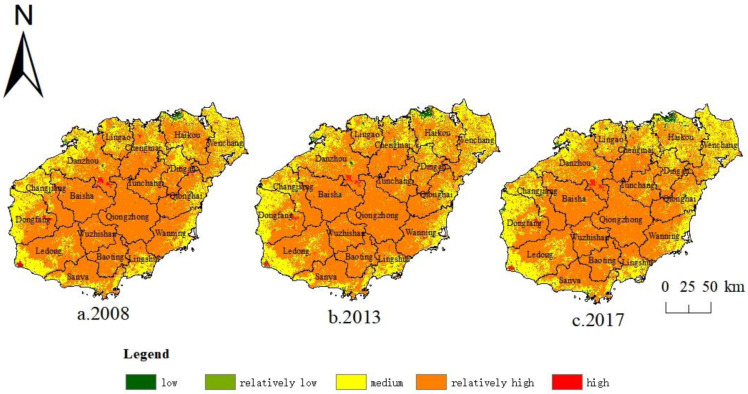
Spatial distribution of ecosystem service values grades on Hainan Island, 2008–2017.

**Table 1 ijerph-20-00776-t001:** Time changes in the ecosystem service values of Hainan Island/10,000 yuan.

Primary Services	Secondary Services	2008	2013	2017
Supply Services	Food production	288,551	201,543	274,915
	Raw material production	248,658	171,341	213,660
	Water supply	−146,593	−103,435	−176,929
	Total value of supply services	390,616	269,449	311,646
	Proportion	3.9%	3.9%	3.7%
Regulation Services	Gas regulation	924,299	638,420	810,563
	Climate regulation	2,407,445	1,658,369	2,062,348
	Purifying the environment	760,017	525,103	653,073
	Water regulation	2,984,338	2,093,598	2,646,761
	Total value of reconciliation services	7,076,099	4,915,490	6,172,745
	Proportion	71.9%	72.1%	72.5%
Support Services	Soil conservation	952,660	655,726	811,387
	Nutrient cycling	96,648	66,904	86,338
	Biodiversity	908,371	626,220	778,739
	Total value of support services	1,957,679	1,348,850	1,676,464
	Proportion	19.9%	19.8%	19.7%
Cultural Services	Aesthetic Landscape	406,310	280,352	348,599
	Proportion	4.1%	4.1%	4.0%
Total		9,830,704	6,814,141	8,509,454

**Table 2 ijerph-20-00776-t002:** Changes in ecosystem service values of various land use types on Hainan Island, 2008–2017.

Land Use Types	Total ESV/Billion			ESV Change/Billion		
	2008	2013	2017	2008–2013	2013–2017	2008–2017
Cultivated land	50.88	35.80	51.59	−15.08	15.79	0.71
Forest	814.35	560.85	693.99	−253.50	133.14	−120.36
Grassland	2.01	0.81	0.80	−1.20	−0.01	−1.21
Water bodies	115.82	83.95	104.56	−31.87	20.61	−11.26
Bare land	0.01	0.00	0.00	−0.01	0.00	−0.01
Total	983.07	681.41	850.94	−301.66	169.53	−132.13

**Table 3 ijerph-20-00776-t003:** Geodetector results.

Detection Factor	NP	PD	LPI	LSI	AREA_MN	CONTAG	IJI	SHDI	SHEI	AI
q-value	0.07	0.09	0.25	0.13	0.41	0.05	0.16	0.36	0.14	0.04
*p*-value	0.36	0.63	0.04	0.15	0.00	0.74	0.68	0.00	0.81	0.16

## Data Availability

Not applicable.

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
