# Peer review of "Dynamic Evolutionary Analysis of Land Use/Cover and Ecosystem Service Values on Hainan Island"

_ijerph, 2022, doi:10.3390/ijerph20010776_

Round 1

Reviewer 1 Report

Analyses of land-use change is an important research issue that indicates the trend of spatial change, which is often the result of spontaneous processes that are difficult to control (especially with inadequate spatial planning policies or economics). Therefore, the topic addressed is relevant, especially in light of the changing development strategy for Hainan Island. It is accepted that land use change is studied over a long time horizon. The authors propose periodisation of the period 2008-2017, which raises questions about the validity of the conclusions drawn. On the plus side, the authors realise that the period adopted is too short. Despite this, I believe that the proposed topic and the methods used make the article interesting and suitable for publication following the introduction/addition of the comments below.

1.      Expand the literature review to include additional items, especially similar studies (using the same, or similar, indicators in recent times).

2.      I propose to make clear the purpose of the article, which will follow from the research questions posed. The objective presented in line 82 seems to go beyond the scope of the article. It is also important to indicate the research gap and to what extent the prepared article fills it.

3.      I propose that the reclassification scheme be switched to the 6 land uses analysed. It is not possible to find out from the text what the source data looks like. It would also be useful to add from what material (satellite images) they were compiled.

4.      Spatial variation is presented only for the ESV indicator. I think it would be useful to also indicate land use changes in the analysed time sections 2008, 2013 and 2017 through maps. The lack of such a study induces a cognitive malaise and hinders the interpretation of the results. As can be seen in the conclusions.

5.      In addition, the variation of the ESV indicator in the analysed years 2008-2017 - according to the adopted classes - is small fig. 11 (the method of dividing the data into classes is not given) so I wonder if presenting this data on maps is necessary?

6.      Chapter 5 should include a discussion of the results obtained with other research results, what is in this section is rather a form of summary. Changing the format of this chapter will also increase the number of items cited by the authors (only 24).

7.      The conclusions do not indicate spatial variation in land use change, which seems natural in this type of study.

Reviewer 2 Report

The authors document and report a thorough study of Land Use/Cover change and Ecosystem Service Values in Hainan Island, based on robust data from 2008, 2013 and 2017.

I believe this is a very well-studied and analyzed report.

Some remarks to address before publishing:

I have difficulty to understand Figure 7. The figure is based on the model by Wang et al. (2022) .

We see in the results Figure 7 that there seems to be practically no Avoidance (Soft green to darker green) or no Tendency (yellow  to red) towards or avoidance of conversion type.

The examples given in Figure 2 are only these above and all other combinations in Figure 7 are not explained. For instance yellow to green? And what do the no color (white) squares represent?

The legends of all the images are too succinct and not self-explanatory.  The legends should explain better what we are seeing.

Fig 10, also needs explanation in the legend. What do the number intervals represent? What ecosystem services are these?

Figure 11. Spatial changes in the ecosystem service values of Hainan Island, 2008-2017. And 2013?

Fig 11 please define these changes in the legend. We only see numbers. What do they mean? How where they calculated? This needs to be in the legends.

Reviewer 3 Report

1. Page 2, lines 78-79, "depict the dynamic changes of land use on the island in relation to "quantity" and "space". What do "quantity" and "space" refer to? This sentence is not clear to me. Please consider revising.

2. The research objective/question is not explicitly written in the article's introduction section. What's the question the research tries to answer?

3. In section 2.2, the data were "uniformly projected to WGS-84". WGS-84 is often considered a geographic coordinate system. Please clarify whether this is the correct projected coordinate system used in this research.

Round 2

Reviewer 1 Report

Previously submitted comments have been included in the text. Thank you for the opportunity to review this interesting text and I wish the authors success.